# Large-Scale Freezing and Thawing Model Experiment and Analysis of Water–Heat Coupling Processes in Agricultural Soils in Cold Regions

Mingwei Hai [1,2,†], Anshuang Su [1,*] , Miao Wang [1,2,3,†], Shijun Gao [1], Chuan Lu [2], Yanxiu Guo [1,2] and Chengyuan Xiao [2]

1   Heilongjiang Province Hydraulic Research Institute, Harbin 150050, China; mingwei_hai@163.com (M.H.); jyllwm1990@126.com (M.W.); youxiang1111219@163.com (S.G.); 15663075573@163.com (Y.G.)
2   School of Architectural and Civil Engineering, Harbin University of Science and Technology, Harbin 150080, China; ludeyouxiang333@163.com (C.L.); 18904604292@163.com (C.X.)
3   Department of Civil Engineering, Harbin Institute of Technology, Harbin 150090, China
*   Correspondence: bridgecrete@163.com
†   These authors contributed equally to this work.

**Abstract:** Heilongjiang Province, the largest commercial grain base in China, experiences significant challenges due to the environmental effects on its soil. The freezing and thawing cycle in this region leads to the transport of water and heat, as well as the exchange and transfer of energy. Consequently, this exacerbates the flooding disaster in spring and severely hampers farming activities such as plowing and sowing. To gain a better understanding of the freezing and thawing mechanisms of farmland soil in cold regions and prevent spring flooding disasters, this study focuses on Heilongjiang Province as a representative area in northeast China. The research specifically investigates the frozen and thawed soil of farmland, using a large-scale low-temperature laboratory to simulate both artificial and natural climate conditions in the cold zone. By employing the similarity principle of geotechnical model testing, the study aims to efficiently simulate the engineering prototypes and replicate the process of large-span and long-time low temperatures. The investigation primarily focuses on the evolution laws of key parameters, such as the temperature field and moisture field of farmland soil during the freeze–thaw cycle. The findings demonstrate that the cooling process of soil can be categorized into three stages: rapid cooling, slow cooling, and freezing stabilization. As the soil depth increases, the variability of the soil temperature gradually diminishes. During the melting stage, the soil's water content exhibits a gradual increase as the temperature rises. The range of water content variation during thawing at depths of 30 cm, 40 cm, 50 cm, and 80 cm is 0.12% to 0.52%, 0.47% to 1.08%, 0.46% to 1.96%, and 0.8% to 3.23%, respectively. To analyze the hydrothermal coupling process of farmland soil during the freeze–thaw cycle, a theoretical model of hydrothermal coupling was developed based on principles of mass conservation, energy conservation, Darcy's law of unsaturated soil water flow, and heat conduction theory. Mathematical transformations were applied after defining the relative degree of saturation and solid–liquid ratio as field functions with respect to the relative degree of saturation and temperature. The simulated temperature and moisture fields align well with the measured data, indicating that the water–heat coupling model established in this study holds significant theoretical and practical value for accurately predicting soil temperature and moisture content during the spring sowing period, as well as for efficiently and effectively utilizing frozen soil resources in cold regions.

**Keywords:** agricultural soils; model test; freeze–thaw cycle; temperature field; water separation field

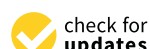



## 1. Introduction

As the northernmost province of China, Heilongjiang Province is influenced by the monsoon climate, and the type of soil present in the region is seasonal frozen soil. Hei-

longjiang Province is also the core area of the Northeast Black Earth Belt, which is one of China's main commodity grain production bases, picking up the national food security "ballast." However, the farmland soil in this region exhibits seasonal freezing and thawing characteristics due to the influence of the freezing and thawing cycles in the seasonal freezing zone. During the freezing period, water accumulates on the surface of the soil, leading to an increase in the water content of the farmland soil. Subsequently, during the spring thawing period, the soil gradually thaws. However, due to the presence of the permafrost layer, the thawed water is unable to infiltrate the ground promptly, resulting in the formation of free water surfaces [1–5]. This presence of free water surfaces exacerbates the occurrence of spring flooding disasters, leading to severe waterlogging on farmland.

During the process of soil freezing, a temperature gradient is established between the colder and warmer ends of the soil as the external temperature decreases. As the temperature in the soil continues to decrease, it eventually reaches a point below the freezing temperature of the water present in the soil. At this point, a portion of the liquid water undergoes a phase transition and transforms into ice. This freezing front then steadily advances from the colder end to the warmer end of the soil. The temperature gradient facilitates the movement of water from the unfrozen region to the frozen region through a thin layer of unfrozen bound water. This transport mechanism allows the release of free energy, which leads to the subcondensation of water into ice. When the spring season brings about an increase in temperature, the soil undergoes a process of absorbing a significant quantity of heat from the surrounding atmosphere. As the soil continues to accumulate heat, reaching a threshold temperature at which the ice within the soil begins to melt, the solid water present in the soil undergoes a gradual transformation into liquid water [6–19]. Hence, it is evident that the soil structure undergoes a close and interactive relationship with water and heat transport during the freezing and thawing processes, resulting in a strong coupling effect that complicates the freezing and thawing processes. Consequently, investigating the patterns of soil moisture and temperature variation in farmland during the freezing and thawing periods holds significant theoretical significance in understanding the occurrence of spring flooding disasters. Moreover, this research can offer initial assurance and technical assistance in preventing the onset of spring flooding disasters in cold regions and maintaining the optimal conditions for crop growth.

Experimental methods are commonly employed to investigate the dynamics of soil moisture and temperature changes during freeze–thaw processes. Sadiq M. F. et al. [20] conducted a study on the effects of temperature gradient and soil thermal properties on freeze swell and water absorption, concluding that these phenomena are influenced by the temperature gradient within the soil. Hou et al. [21] examined the impacts of freezing temperature, initial moisture content, and dry density on soil water–heat transport through indoor soil one-way freezing tests. Wang et al. [22] performed one-dimensional freezing tests on Qinghai-Tibetan chalky clay soils with varying initial water contents in an open system, utilizing CCD image acquisition and CT scanning techniques. Their findings indicated that the freezing depth increased with increasing initial water content, step-like freezing structures appeared in the vertical profile, more structural polygons were observed in the horizontal profile, and ice lens segregation, solidification of unfrozen zones, and water migration during freezing were more pronounced. Bai et al. [23] investigated the water-vapor-heat behavior in frozen unsaturated coarse-grained soils and found that vapor migration was significant when the initial moisture content of the soil was low and water migration was dominant when the initial moisture content of the soil was high. Li et al. [24] conducted one-dimensional freezing experiments in an open system and found that the total amount of freezing swell was proportional to the amount of water uptake, and furthermore, the freezing swell increased linearly with the increase in the fines content. Gao et al. [25] conducted a series of one-way freezing experiments in an open system with an unpressurized water supply and concluded that fine-grain content and water content are the main factors affecting the frost susceptibility of coarse-grained soils. However, investigating the mechanism and change process of soil hydrothermal interactions solely

through indoor experimental methods poses several challenges in the complex and variable system of agricultural soils. It is difficult to accurately simulate the actual environment, reflect the true scale of agricultural soils, and control external influences. Large-scale geotechnical modeling tests, based on similarity theory, have become an indispensable tool for studying permafrost mechanics [26,27]. These tests can more accurately replicate the actual environmental conditions during the freezing and thawing of agricultural soils, reduce the research cycle, better illustrate the structural changes of agricultural soils during freezing and thawing, and provide a more realistic representation of the temperature and moisture field changes. However, large-scale geotechnical modeling tests are rarely applied to farmland soil backgrounds in current research. Furthermore, advancements in modeling technology have led to the development of various models specifically designed to address soil freeze–thaw phenomena, such as the SHAW model, CLASS, FROST, and SEWAB [28–32]. Many of these models are complex and time-consuming to simulate, posing a challenge to constructing faster and more efficient numerical models.

In this research, the study area chosen was Heilongjiang Province, which is a representative cold region in northeast China. The focus of the study was on the freeze–thaw soil of farmland, which served as the subject of investigation. The objective was to examine the changes in temperature and moisture within the farmland soil during the freeze–thaw cycle. This was achieved by artificially simulating the climate conditions of the cold region and natural climate conditions in a large-scale low-temperature laboratory. The study employed principles such as mass conservation, energy conservation, Darcy's law of unsaturated soil water flow, and the theory of heat conduction. By defining the relative saturation and the solid–liquid ratio and utilizing mathematical transformations, a theoretical model of soil water–heat coupling was constructed. This model incorporated relative saturation and temperature as the field functions. The analysis of the water–heat coupling process in farmland soil during the freeze–thaw cycle, accurate prediction of soil temperature and moisture content during the spring sowing period, and efficient and rational utilization of frozen soil resources in cold regions hold significant theoretical value and practical importance.

## 2. Materials and Methods

### 2.1. Similarity Scale for Geotechnical Modeling Tests

The similarity criterion holds significant importance in the field of geotechnical modeling test research, with particular emphasis on the similarity ratio scale problem in permafrost modeling test research. To ensure the accuracy and reliability of the model, it is crucial to establish the appropriate ratio scale relationships between the prototype and the model. In this section of the model test, the determination of these specific scale relationships was achieved by conducting a thorough review of relevant references [26,27]. The specific scale relationships obtained are presented in Table 1. The similarity criterion is the basic criterion in the experimental study of the geotechnical model. N and 1 indicate the scale relationship between the prototype and the model (the ratio of the prototype to the model), and 1 indicates that the scale of the prototype and the model is 1, which means that the density, cohesion, angle of internal friction, temperature, heat diffusion coefficient, thermal conductivity, and pore water pressure of the prototype and the model are considered to be the same in the experimental process.

**Table 1.** Similarity criteria in geotechnical modeling tests.

| Physical Quantity | Model Scale (Ratio of Prototype to Model) |
|---|---|
| lengths | N |
| Density | 1 |
| Cohesion | 1 |
| Angle of internal friction | 1 |
| Temperature | 1 |
| Thermal Diffusion Coefficient | 1 |
| Thermal conductivity | 1 |

**Table 1.** *Cont.*

| Physical Quantity | Model Scale (Ratio of Prototype to Model) |
|---|---|
| Pore water pressure | 1 |
| Time (unfrozen water migration) | N2 |
| Time (heat exchange) | N2 |

*2.2. Experiment Design*

2.2.1. Test Equipment

This experiment is conducted at the Wanjia Experimental Station of the Heilongjiang Water Resources Research Institute, specifically in their low-temperature laboratory. The laboratory houses an indoor model test chamber with dimensions of 3.0 m × 1.0 m × 1.3 m (L × W × H). The temperature within the laboratory is regulated by a temperature control system that can both warm and cool the environment. The schematic diagram of the model box and temperature control equipment can be seen in Figure 1. In order to collect data on the temperature and moisture of the soil during freeze–thaw cycles, several instruments are utilized. These instruments include the Cold Zone Permafrost Laboratory Monitoring System, DT80 Data Acquisition Instrument Mainframe + CEM20 Expansion Module (CAS Dataloggers, Chesterfield, OH, USA), PT100 Temperature Sensor (IST, Ebnat-Kappel, Switzerland), and TDR-3 Moisture Sensor (Nutech International, New Delhi, India), as depicted in Figure 2.

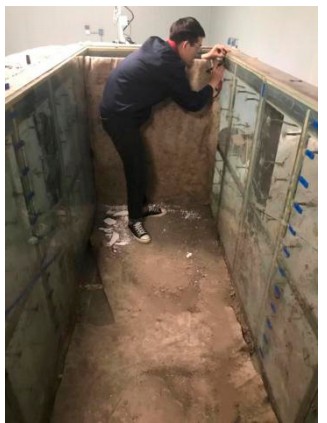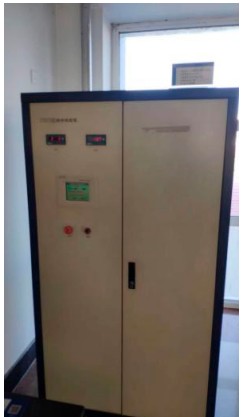

**Figure 1.** Schematic diagram of low-temperature laboratory box and model test temperature control system.

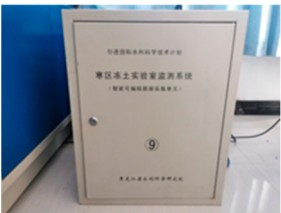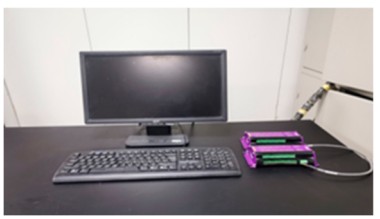

(**a**) Observation system　　　　　　(**b**) DT80 Data acquisition instrument

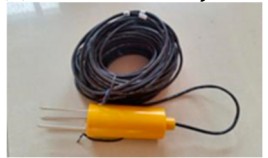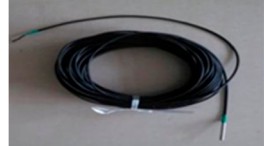

(**c**) TDR-3 Moisture sensor　　　　　　(**d**) PT100 Temperature sensor

**Figure 2.** Temperature control systems.

### 2.2.2. Model Design and Production

The experiment involved the utilization of powdery clay as the foundation soil. The soil samples were obtained by sun-drying and crushing the clay, followed by sieving through a 2 mm sieve. The samples were then prepared at a moisture content of 30% and a controlled dry density of 1.65 g/cm³. To ensure uniform moisture distribution, the surface of the soil samples was covered with cling film and left undisturbed for 48 h. It should be noted that the soil was not compacted to its maximum capacity, resulting in a general degree of compaction of 95%. In order to achieve uniform soil distribution, the filling process was conducted in layers, with each layer being 10 cm thick. The natural dry density of the soil served as the quality control standard for the layer-by-layer filling. Throughout the filling process, strict control was maintained over the height of the soil model. The completed soil model can be observed in Figure 3. To ensure the accuracy of the experimental results, three compartments were set up for parallel experiments. In each compartment, exactly the same experimental operations were performed to minimize random errors and uncertainties. In this way, the reliability and reproducibility of the experimental results can be better determined.

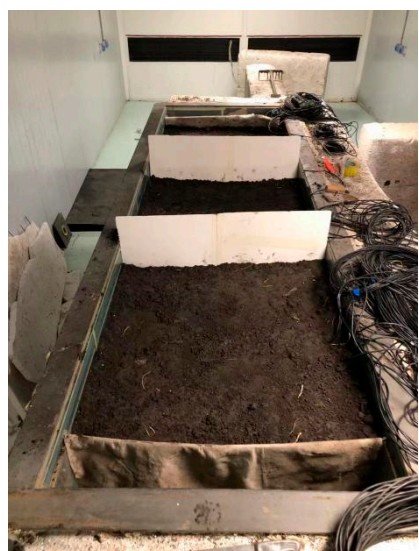

**Figure 3.** Model Preparation Diagram.

### 2.2.3. Measuring Instrument Arrangement

Based on the scale and dimensions of the model test, as well as the soil density in the field test section of Wanjia Experimental Station of the Heilongjiang Provincial Institute of Water Resources Science, a model was constructed to simulate the freeze–thaw cycling process of the soil indoors. This model was designed to monitor the temperature and moisture variations during the freeze–thaw cycle, with the aim of providing data and references for numerical simulations.

(1) Temperature field monitoring

The temperature field monitoring primarily focuses on monitoring the ambient temperature within the model test box and the temperature of the soil body contained within the test box. The ambient temperature plays a crucial role in influencing the temperature of the soil. To effectively monitor the changes in ambient temperature, it is necessary to identify a temperature uniformity point where temperature sensors can be strategically placed above the indoor model. In terms of soil temperature monitoring, a total of 33 monitoring points were established within the model test. These monitoring points were divided into three groups: points 1 to 11 were designated for monitoring the temperature of the first soil body, points 12 to 22 for the second soil body, and points 22 to 32 for the third soil body. Within the depth range of 0 to 80 cm, temperature sensors were arranged

at intervals of 10 cm, while within the depth range of 80 to 120 cm, sensors were placed at intervals of 20 cm. The specific arrangement of temperature sensors can be observed in Figure 4.

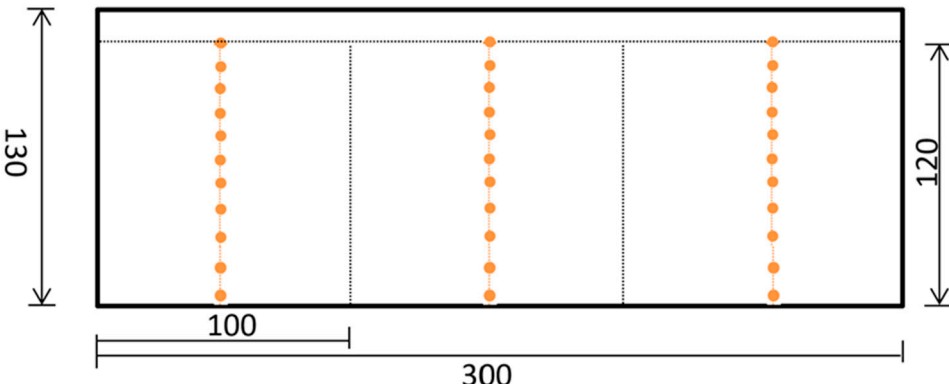

**Figure 4.** Temperature sensor layout.

(2) Layered water content monitoring

The monitoring of layered water content serves as an indicator of the water migration within the soil body during the freeze–thaw cycle. This test is conducted both prior to the commencement and after the completion of the experiment. Before the test begins and after it concludes, water observations are conducted on different sections of the soil body. The primary method involves extracting samples at 10 cm intervals from each test hole to observe the moisture content at various layers. As the sampling process involves creating boreholes, it is necessary to refill the boreholes with residual soil to prevent any interference with the temperature field observations. Additionally, three moisture sensors were installed in the surface layer of each soil for comparative purposes. Within the depth range of 0 to 40 cm, one sensor was placed at intervals of 20 cm. The moisture sensor arrangement is shown in Figure 5.

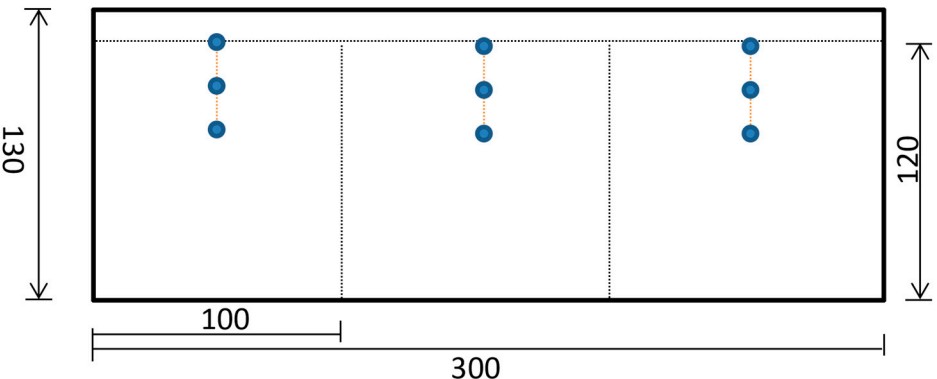

**Figure 5.** Moisture sensor layout.

2.2.4. Meteorological Conditions in the Field Test Section

The temperature of the test environment was regulated based on observations from the Harbin field permafrost observation field. Historical data spanning several years indicated that temperatures in the region could reach as low as −32 °C to −37 °C, with an average annual temperature ranging from 2 °C to 4 °C. The area experiences approximately 2800 h of sunlight per year, with the longest duration occurring during the growing season (May to September) in Heilongjiang Province. During the indoor freezing and thawing model tests, the daily sunlight exposure ranges from 1217 h to 1374 h. The first occurrence of frost typically transpires in mid-September, and the frost-free period lasts approximately

120 to 150 days annually. The maximum depth of freezing ranges from 2.0 m to 2.9 m, with a freezing period lasting over 5 months, accounting for half of the year. In summary, the field test section is characterized by extended periods of sunlight, high cumulative temperatures, rapid cooling, an early onset of frost, a prolonged freezing period, and low freezing temperatures. The temperature fluctuations in the soil at the field test site are primarily influenced by the climatic conditions, necessitating the simulation of the field's temperature environment and effective control of ambient temperature during indoor model tests to achieve comparable results.

### 2.2.5. Temperature Control Program for Indoor Model Tests

For the design of the experimental temperature regime, we referred to the 2020–2021 daily average air temperature change data from the Harbin Field Permafrost Observation Field. The highest temperature occurs in early July, with a maximum of 27.3 °C, and the lowest temperature occurrs around January, with a minimum of −28.5 °C. Temperatures entered a negative phase in mid-November and gradually declined, and the soil began to freeze downward. Temperatures began to rise in early February and entered a constant positive temperature phase in early April, when the topsoil began to gradually thaw downward. The soil body completes a freeze–thaw cycle from the beginning of freezing to the complete thawing of the soil body.

The freeze–thaw cycle was divided into three distinct phases, each requiring specific temperature control measures in order to accurately simulate the conditions in an indoor model test. The first phase, known as the rapid cooling phase, lasted for 13 h (from 0 h to 13 h) and involved controlling the ambient temperature from 10.0 °C to −24.0 °C. This phase aimed to replicate the transition from the relatively warmer fall season to the colder winter months, during which the soil begins to freeze. The second phase, referred to as the cold stabilization phase, spanned 293 h (from 13 h to 306 h) and maintained the ambient temperature at −24.0 °C. This phase aimed to simulate the stabilization of winter temperatures at a lower level and the subsequent intensification of soil freezing until the maximum depth of freeze was achieved. Lastly, the warming phase lasted for 960 h (from 330 h to 1290 h) and involved controlling the ambient temperature within the range of −24.0 °C to 20.0 °C. This phase aimed to replicate the gradual increase in ambient temperature during the arrival of spring, leading to the thawing of the soil until it was completely thawed, thus completing the freeze–thaw cycle.

## 3. Test Results and Analysis

### 3.1. Temperature Development of the Samples

Figure 6a illustrates the change in soil temperature over time during the freezing process. In the illustration, 0 cm represents the top layer of soil, 10 cm represents 10 cm from the top layer of soil, and 80 cm represents 80 cm from the top layer of soil. The process can be divided into three main stages: rapid cooling, slow cooling, and freezing stabilization. The rapid cooling stage, which lasts from 0 to 58 h, is characterized by a significant decrease in soil temperature. The temperature differences at various depths are as follows: 31.957 °C at 10 cm, 28.833 °C at 20 cm, 21.088 °C at 30 cm, 16.215 °C at 40 cm, 12.963 °C at 50 cm, 12.963 °C at 60 cm, and 16.215 °C at 60 cm. The slope of the soil temperature versus time curve shows that energy exchange and temperature changes are more pronounced at shallower depths. As the depth increases, the amplitude of temperature change decreases, indicating a gradual reduction in temperature volatility. The decrease in the cold end temperature leads to a downward progression of the negative temperature and an increase in the temperature gradient. Consequently, the freezing front moves rapidly downward, resulting in relatively smooth temperature changes in the deeper soil layers and violent temperature changes in the surface layer. The slow cooling phase is a stage of soil temperature change that may not be very obvious at the beginning of the process but can be initially judged by observing the trend of soil temperature change. When the rate of temperature decline gradually slows down, this may indicate that the slow

cooling phase has begun. After 58 h, the rate of temperature drop gradually slows down, marking the onset of the slow cooling stage. Eventually, as time progresses, the temperature stabilizes and remains relatively constant, indicating the freezing stabilization stage.

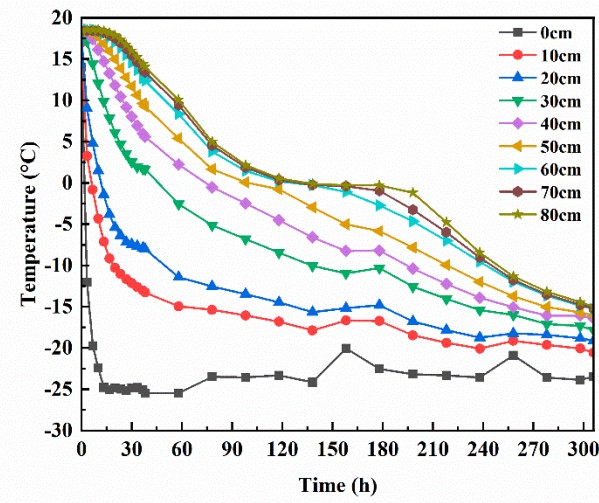

(**a**) Curve of soil temperature variation with time during freezing process

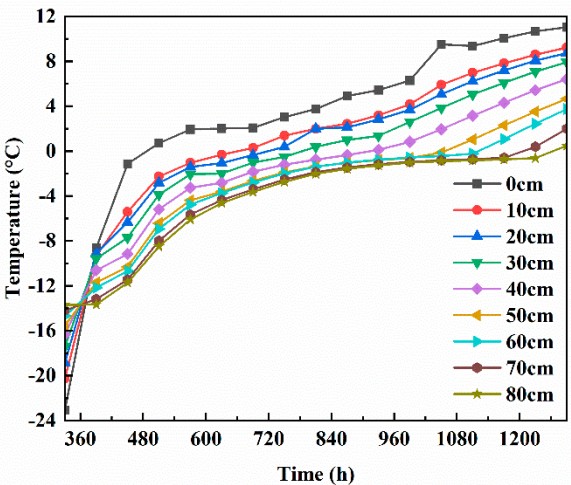

(**b**) Curve of soil temperature variation with time during melting process

**Figure 6.** Temperature–time curve of soil samples at different positions.

Figure 6b illustrates the temporal change in soil temperature throughout the entire melting process. During the melting stage, the soil temperature exhibited a gradual increase. Notably, there were significant variations in soil temperature changes across different soil layers. Specifically, the soil temperature at a depth of 10 cm experienced a change of 28.64 °C, while the temperature change at a depth of 20 cm was 28.69 °C. At a 30 cm depth, the temperature change was 25.71 °C, followed by 23.5 °C at a 40 cm depth, 22.09 °C at a 50 cm depth, 20.39 °C at a 60 cm depth, 18.93 °C at a 70 cm depth, and 17.35 °C at a 80 cm depth. As the soil layer depth increased, the amplitude of soil temperature change sequentially decreased, indicating a more stable trend in temperature variation. This can be attributed to the fact that deeper soil layers are further insulated from external temperatures, resulting in greater heat loss through the process of heat conduction. Consequently, the amplitude of temperature change in this region, relative to the shallow soil area, exhibits a decreasing trend and a certain delay phenomenon.

As depicted in Figure 6, the freezing stage is characterized by the downward movement of the freezing line, represented by the 0 °C temperature contour, as the external

ambient temperature gradually decreases. This movement signifies an increase in the depth of freezing. Conversely, during the thawing stage, the upper surface temperature of the soil gradually rises as the external ambient temperature increases. This results in the gradual downward movement of the 0 °C isotherm, indicating an expansion in the depth of thawing.

### 3.2. Measurement of Sample Water Content

The phenomenon of soil freezing and thawing primarily involves the phase change of water. During soil freezing, when the upper surface is subjected to low-temperature boundary conditions, the temperature of the upper soil gradually decreases, resulting in the formation of a temperature gradient within the soil. As the temperature of the upper soil drops below the phase transition temperature of pore water, ice begins to form between the soil pores. This formation of pore ice, along with the temperature gradient, causes the unfrozen pore water to gradually migrate towards the frozen area, resulting in a change in the distribution of pore water content. Conversely, during soil thawing, when the upper surface is subjected to high-temperature boundary conditions, the temperature of the upper soil gradually increases. The pore ice in the soil begins to melt from the top downward, while the deeper layers of soil still retain frozen pore ice. This leads to a blockage of the melted pore water in the upper soil, causing an increase in the pore water content in that region as the pore ice continues to melt.

Figure 7 depicts the moisture change patterns during the soil melting stage. The figures illustrate that as the temperature rises, the soil moisture undergoes a transition from solid to liquid, resulting in a gradual increase in soil water content. Specifically, at a depth of 30 cm, the water content varies between 0.12% and 0.52% during the melting process. At a depth of 40 cm, the water content ranges from 0.47% to 1.08%. Similarly, at a depth of 50 cm, the water content varies between 0.46% and 1.96%, while at a depth of 80 cm, the water content ranges from 0.8% to 3.23%. During soil thawing, water in the upper thaw zone moves downwards towards the freeze–thaw interface due to the combined effects of gravity and temperature gradient. Some of this water freezes into ice, while the remaining portion exists in liquid form. However, the presence of the aggregated ice layer hinders further infiltration, causing the water from the thaw zone to accumulate near the freeze–thaw interface, resulting in the aggregation of liquid water.

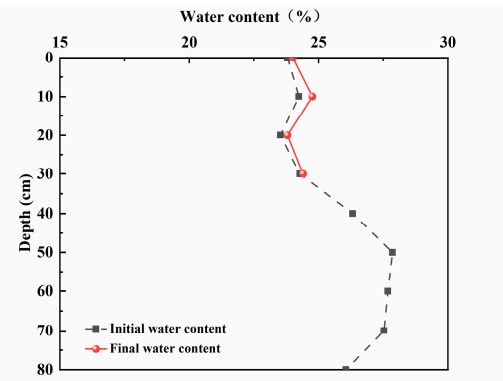

(**a**) Liquid water content at 30 cm

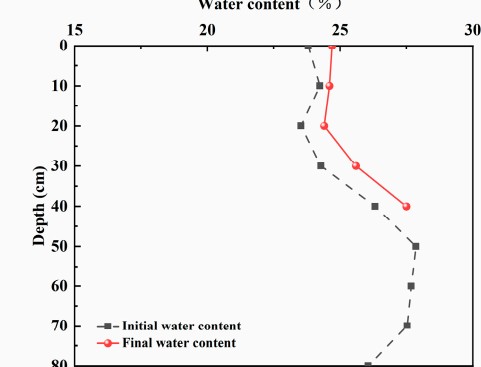

(**b**) Liquid water content at 40 cm

**Figure 7.** *Cont.*

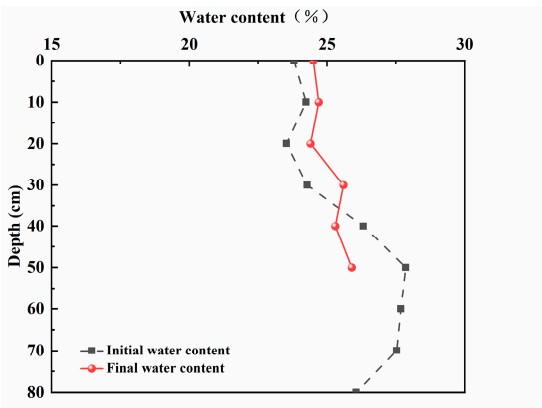
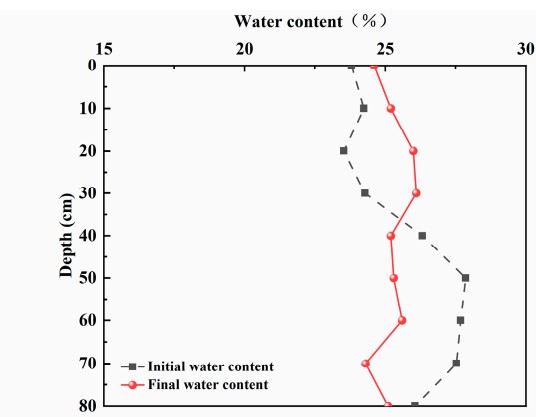

(**c**) Liquid water content at 50 cm          (**d**) Liquid water content at 80 cm

**Figure 7.** Moisture change curve in soil melting stage.

## 4. Frozen Soil Hydrothermal Coupling Equation

### 4.1. Model Assumptions

In scenarios characterized by limited interaction between water and heat, the challenge of water-thermal coupling can be resolved by initially solving the moisture equation of motion, followed by the heat flow migration equation. When dealing with soil in areas subject to seasonal freezing, the existence of temperature disparities leads to moisture redistribution, thereby making the water-thermal interaction significant. To effectively tackle these concerns, an extra linkage equation must be incorporated to address the water–heat coupling problem, in conjunction with the moisture motion equation and the heat flow migration equation. The proposed model is founded upon the following assumptions:

(1) Moisture migration is observed in the form of liquid water and follows the generalized Darcy's law, which describes the flow of fluids through porous media.
(2) The presence of gaseous water and its transformation into liquid water are not taken into account in this study.
(3) The influence of salts and mineral ions on the migration of water is not considered in the analysis.
(4) The soil is assumed to be an isotropic material, meaning that its heat transfer properties are uniform in all directions.
(5) The calculation assumes that there is no loss of temperature and that the water content and temperature in the permafrost are in equilibrium.
(6) The soil particles, liquid water, and ice are assumed to be incompressible, meaning that their volume does not change under pressure.

### 4.2. Moisture Equation of Motion

Considering that the unsaturated soil mass experiences freezing and consists of three-phase constituents, namely soil particles, unfrozen water, and ice, the primary area of interest for examination is a distinct component within the soil mass, as illustrated in Figure 8.

According to the law of conservation of mass, the decrease in water diversion into the cell at any given time is equal to its increase. In time dt:

$$\Delta m_d = \Delta m_u \tag{1}$$

where $\Delta m_d$ denotes the decrease in mass of water inflow into the unit in dt time, kg; $\Delta m_u$ denotes the increase in mass of the unit in dt time, kg.

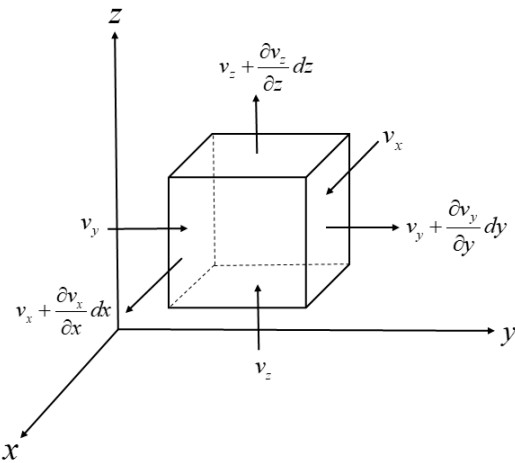

**Figure 8.** Schematic diagram of water migration in the soil microfraction unit.

The reduction in the inflow of water into the unit at time dt is the sum of the reduction in the inflow of water into the unit in the three directions, as shown in the following equation:

$$\Delta m_d = \Delta m_{dx} + \Delta m_{dy} + \Delta m_{dz} \tag{2}$$

where $\Delta m_{dx}$, $\Delta m_{dy}$, and $\Delta m_{dz}$ denote the mass reduction of water inflow into the unit in the $x$, $y$, and $z$ directions, respectively, in dt time, kg.

$$\Delta m_{dx} = \rho_w v_x dy dz dt - \rho_w (v_x + \frac{\partial v_x}{\partial x} dx) dy dz dt = -\rho_w \frac{\partial v_x}{\partial x} dx dy dz dt \tag{3}$$

$$\Delta m_{dy} = \rho_w v_y dx dz dt - \rho_w (v_y + \frac{\partial v_y}{\partial y} dy) dx dz dt = -\rho_w \frac{\partial v_y}{\partial y} dx dy dz dt \tag{4}$$

$$\Delta m_{dz} = \rho_w v_z dx dy dt - \rho_w (v_z + \frac{\partial v_z}{\partial z} dz) dx dy dt = -\rho_w \frac{\partial v_z}{\partial z} dx dy dz dt \tag{5}$$

where $v_x$, $v_y$, and $v_z$ are the components of the moisture flux $v$ in the $x$, $y$, and $z$ directions, m/s.

Summing the three components yields the following equation:

$$\Delta m_d = -\rho_w \frac{\partial v_x}{\partial x} dx dy dz dt - \rho_w \frac{\partial v_y}{\partial y} dx dy dz dt - \rho_w \frac{\partial v_z}{\partial z} dx dy dz dt \tag{6}$$

To simplify the calculation, the Hamiltonian operator is introduced:

$$\Delta m_d = -\rho_w \nabla(v) dx dy dz dt \tag{7}$$

The mass increase in water in the unit during time dt is given in the following equation:

$$\Delta m_u = \frac{\partial \theta}{\partial t} \rho_w dx dy dz dt \tag{8}$$

The water mass inside the unit is considered to consist of unfrozen water and ice, and the volume content of unfrozen water inside the soil unit is $\theta_u$ (%) and the volume content of ice is $\theta_i$ (%). The density of ice is $\rho_i$, which is given in the following equation:

$$\theta = \theta_u + \frac{\rho_i}{\rho_w} \theta_i \tag{9}$$

The mass increase in the unitary water in time dt can be replaced by the following equation:

$$\Delta m_u = \frac{\partial \theta}{\partial t}\rho_w dxdydzdt = \frac{\partial \theta_u}{\partial t}\rho_w dxdydzdt + \frac{\rho_i}{\rho_w}\frac{\partial \theta_i}{\partial t}\rho_w dxdydzdt \tag{10}$$

Substituting Equations (7) and (10) into Equation (1) and eliminating the same terms on both sides of the equation, we can obtain the water migration equation:

$$\frac{\partial \theta_u}{\partial t} + \frac{\rho_i}{\rho_w}\frac{\partial \theta_i}{\partial t} + \nabla(v) = 0 \tag{11}$$

According to Richard's equation [33] and considering the hindering effect of pore ice on unfrozen water migration [34], the differential equation for unfrozen water migration in unsaturated frozen soil is:

$$\frac{\partial \theta_u}{\partial t} + \frac{\rho_i}{\rho_w}\frac{\partial \theta_i}{\partial t} = \nabla[D(\theta_u)\nabla\theta_u + k(\theta_u)] \tag{12}$$

where $\theta_u$ is the volume content of unfrozen water in permafrost; $k(\theta_u)$ is the permeability coefficient of unsaturated soil, m/s; and $D(\theta_u)$ is the diffusivity of water in permafrost, $m^2/s$. The diffusivity of water in permafrost is calculated as follows [34]:

$$D(\theta_u) = \frac{k(\theta_u)}{c(\theta_u)} \cdot I \tag{13}$$

where $I$ is the impedance factor [34], which represents the retarding effect of pore ice on unfrozen water migration and is calculated by the following equation:

$$I = 10^{-10\theta_i} \tag{14}$$

### 4.3. Heat Flow Migration Equation

When the temperature of the soil drops below the phase change temperature, a morphological transformation occurs in the unfrozen water present in permafrost. This transformation results in the release of a substantial amount of heat, which acts as an internal heat source. To facilitate calculations, it is assumed that the thermal energy gained by the frozen soil (Q) is equivalent to the sum of the net heat transfer ($Q_1$) and the latent heat of the phase change of ice and water ($Q_2$) [35]. Equation (15) can be derived based on this assumption.

$$Q = Q_1 + Q_2 \tag{15}$$

Still analyzing the cell shown in Figure 9, the heat $Q_1$ that is transferred into the soil cell at any time dt is expressed as

$$Q_1 = Q_x + Q_y + Q_z \tag{16}$$

where $Q_x$, $Q_y$, and $Q_z$ are the components of the heat transferred into the soil unit in the x, y, and z directions, respectively.

$$Q_x = q_x dydzdt - (q_x + \frac{\partial q_x}{\partial x}dx)dydzdt = -\frac{\partial q_x}{\partial x}dxdydzdt \tag{17}$$

$$Q_y = q_y dxdzdt - (q_y + \frac{\partial q_y}{\partial y}dy)dxdzdt = -\frac{\partial q_y}{\partial y}dxdydzdt \tag{18}$$

$$Q_z = q_z dxdydt - (q_z + \frac{\partial q_z}{\partial z}dz)dxdydt = -\frac{\partial q_z}{\partial z}dxdydzdt \tag{19}$$

where, $q_x$, $q_y$, and $q_z$ are the components of heat flow density $q$ in the $x$, $y$, and $z$ directions, $W/m^2$.

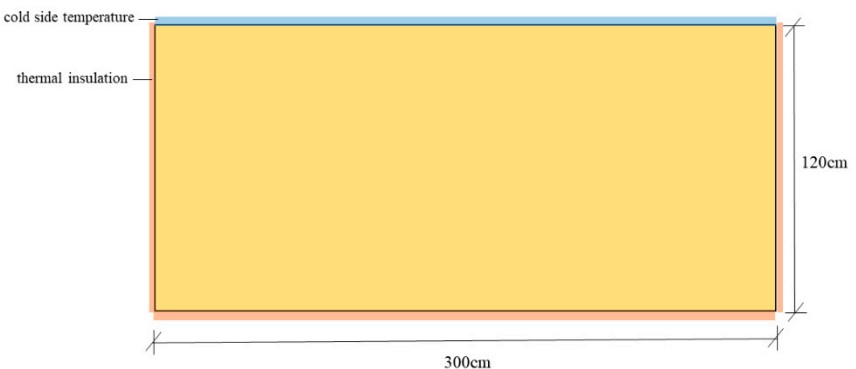

**Figure 9.** Schematic diagram of soil calculation model.

For the heat transfer problem, the following equation is obtained from Fourier's law:

$$q = -\lambda \nabla T \tag{20}$$

where $\lambda$ is the thermal conductivity, W/(m·°C).

Thus, Equation (16) can be replaced with the following equation:

$$Q_1 = -\frac{\partial q_x}{\partial x}dxdydzdt - \frac{\partial q_y}{\partial y}dxdydzdt - \frac{\partial q_z}{\partial z}dxdydzdt = \nabla(\lambda\nabla T)dxdydzdt \tag{21}$$

At any time dt, the soil undergoes a temperature change $Q$ expressed as the following equation:

$$Q = C\frac{\partial T}{\partial t}dxdydzdt \tag{22}$$

where C is the volumetric heat capacity, J/(m$^3$·°C); T is the temperature value, °C; t is the time, s.

Considering the phase change effect of the soil freezing process, the latent heat of phase change $Q_2$ can be expressed as

$$Q_2 = L\rho_i\frac{\partial\theta_i}{\partial t}dxdydzdt \tag{23}$$

where L is the latent heat of phase change, kJ/kg.

Substituting Equations (21)–(23) into Equation (15), the heat transfer equation is obtained [36].

$$\rho C(\theta)\frac{\partial T}{\partial t} = \nabla[\lambda(\theta)\nabla T] + L \cdot \rho_i\frac{\partial\theta_i}{\partial t} \tag{24}$$

The specific heat of a soil can be expressed as a volume-weighted average of the various material components, while the thermal conductivity has an exponentially weighted nature. The volumetric specific heat C(θ) and the thermal conductivity λ(θ) can be described as [37].

$$C(\theta) = C_s \cdot \rho \cdot \theta_s + C_i \cdot \rho_i \cdot \theta_i + C_w \cdot \rho_w \cdot \theta_u \tag{25}$$

$$\lambda(\theta) = \lambda_s^{\theta_s} \cdot \lambda_i^{\theta_i} \cdot \lambda_w^{\theta_u} \tag{26}$$

where T is the instantaneous temperature of the soil, °C; t is the time, s; θ is the volumetric water content, and $\theta_i$ is the pore ice volume content; $\rho$, $\rho_w$, and $\rho_i$ are the densities of soil, water, and ice, respectively, kg/m$^3$; $\lambda_s$, $\lambda_w$, and $\lambda_i$ are the thermal conductivity of soil, water, and ice, respectively, W/(m·°C); $C_s$, $C_w$, and $C_i$ are the specific heat capacities of soil, water, and ice, respectively, J/(m$^3$·°C).

### 4.4. Phase Change Dynamic Equilibrium Relationship

According to the control equation governing the hydrothermal coupling of frozen soil, there are three variables that are not known: temperature, volume content of pore ice, and volume content of unfrozen water. To solve this equation, it is imperative to introduce a linkage equation that encompasses these three variables. This linkage equation is essential in order to obtain a comprehensive solution for the control equation, thereby establishing the relationship between the variables $\theta_i$, $\theta_u$, and $T$.

In this paper, using the solid–liquid ratio (the ratio of the volume of pore ice to the volume of unfrozen water in permafrost), denoted as $B_I$ [38], it is shown that:

$$B_I(T) = \frac{\theta_i}{\theta_u} = \begin{cases} 1.1 \left( \frac{T}{T_f} \right)^B - 1.1 & T < T_f \\ 0 & T \geq T_f \end{cases} \tag{27}$$

where $T_f$ is the freezing temperature of the soil; B is the solid–liquid ratio coefficient.

### 4.5. Theoretical Model of Water–Heat Coupling in Frozen Soil Based on Relative Saturation

If we define the relative saturation, S, as:

$$S = \frac{\theta_u - \theta_r}{\theta_s - \theta_r} \tag{28}$$

Then the permeability of unsaturated soils and specific water capacity can be expressed as [39,40]:

$$k(\theta_u) = k_s \cdot S^l \left( 1 - \left( 1 - S^{1/m} \right)^m \right)^2 \tag{29}$$

$$c(\theta_u) = a_0 m / (1 - m) \cdot (\theta_s - \theta_r) \cdot S^{1/m} \left( 1 - S^{1/m} \right)^m \tag{30}$$

where $\theta_s$ and $\theta_r$ represent the saturated water content and residual water content of the soil, respectively.

Therefore, the equation of water movement can be expressed as the following equation:

$$\frac{\partial S}{\partial t} + \frac{\rho_i}{\rho_w} \cdot \left[ \left( \frac{\partial B_I(T)}{\partial t} \cdot S + B_I(T) \cdot \frac{\partial S}{\partial t} \right) + \frac{\theta_r}{(\theta_s - \theta_r)} \frac{\partial B_I(T)}{\partial t} \right] = \nabla [D(S) \nabla S + k(S)] \tag{31}$$

The heat flow migration equation can be expressed as the following equation:

$$\rho C(\theta) \frac{\partial T}{\partial t} + \nabla \cdot (-\lambda(\theta) \nabla T) = L \cdot \rho_i \cdot \left[ (\theta_s - \theta_r) \cdot \left( \frac{\partial B_I(T)}{\partial T} \frac{\partial T}{\partial t} \cdot S + B_I(T) \cdot \frac{\partial S}{\partial t} \right) + \theta_r \frac{\partial B_I(T)}{\partial T} \frac{\partial T}{\partial t} \right] \tag{32}$$

### 4.6. Model Validation

The computational model depicted in Figure 9 exhibits the same dimensions as the model test soil in terms of length and height. The boundary conditions and initial conditions employed in the model are consistent with those used in the test. The horizontal displacements on both sides of the soil sample and the vertical displacements at the bottom boundary are constrained. The thermodynamic parameters of the soil samples utilized in this calculation were obtained from relevant literature sources [41,42]. Similarly, the hydrodynamic parameters were obtained from literature references [41,42]. Table 2 presents the thermodynamic and hydromechanical parameters necessary for analyzing the soil hydrothermal coupling model. The COMSOL platform was employed for the secondary development of the derived hydrothermal coupling model, and numerical simulations were conducted to assess the model's performance under various working conditions.

**Table 2.** Computed parameters.

| Parameter | Value | Unit | Parameter | Value | Unit |
|---|---|---|---|---|---|
| $C_s$ | $0.84 \times 10^3$ | $J/(kg \cdot {}^\circ C)$ | $\rho_s$ | $1.9 \times 10^3$ | $kg/m^3$ |
| $C_w$ | $4.180 \times 10^3$ | $J/(kg \cdot {}^\circ C)$ | $\theta_s$ | 0.5 | — |
| $C_i$ | $2.100 \times 10^3$ | $J/(kg \cdot {}^\circ C)$ | $\theta_r$ | 0.02 | — |
| $\lambda_s$ | 1.3 | $W/(m \cdot {}^\circ C)$ | $T_f$ | $-0.54$ | ${}^\circ C$ |
| $\lambda_w$ | 0.63 | $W/(m \cdot {}^\circ C)$ | $B$ | 0.56 | — |
| $\lambda_i$ | 2.31 | $W/(m \cdot {}^\circ C)$ | $k_s$ | $0.96 \times 10^{-7}$ | $m/s$ |
| $L$ | $3.3456 \times 10^5$ | $J/kg$ | $l$ | 0.5 | — |
| $\rho_i$ | $0.918 \times 10^3$ | $kg/m^3$ | $a$ | 2.59 | — |
| $\rho_w$ | $1.000 \times 10^3$ | $kg/m^3$ | $m$ | 0.22 | — |

Figure 10 illustrates the patterns of temperature variation at nine specific observation points in the specimen, both measured and numerically calculated. Generally, the numerically calculated temperature changes align well with the measured data. During the freezing stage, the temperatures at the observation points deviate from the initial temperature and gradually stabilize over time, reaching a constant value. The freezing process is characterized by the downward movement of the freezing surface and an increase in the depth of the frozen area. The advancement of the freezing front can be divided into three stages. In the initial 58 h, the freezing front progresses rapidly, accompanied by a larger temperature gradient. Moisture begins to migrate from the bottom to the top, resulting in an increase in water content at the top. After 58 h, the speed of the freezing front gradually slows down, leading to a slower temperature change. This allows sufficient time for the bottom water to migrate upwards. Eventually, the freezing front ceases to advance, and the temperature field stabilizes. During the melting phase, the soil temperature gradually rises and exhibits significant variation between layers. Furthermore, it is evident that in the early stages of thawing, the points closer to the upper boundary of the soil column experience a faster rate of temperature increase. This can be attributed to the larger temperature gradient near the upper interface of the soil column.

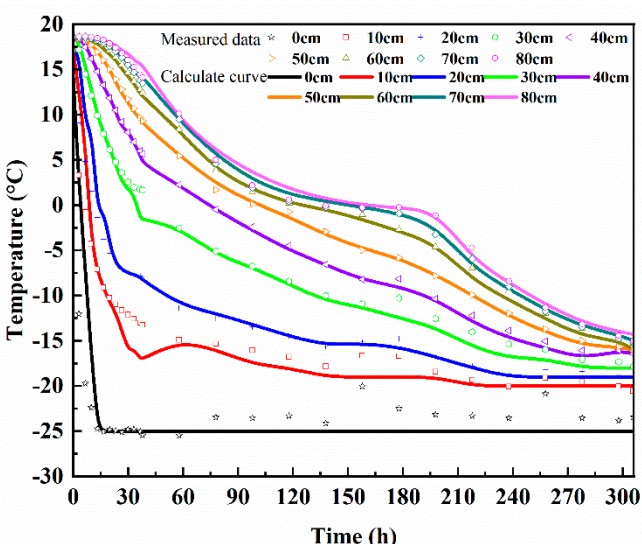

**Figure 10.** *Cont.*

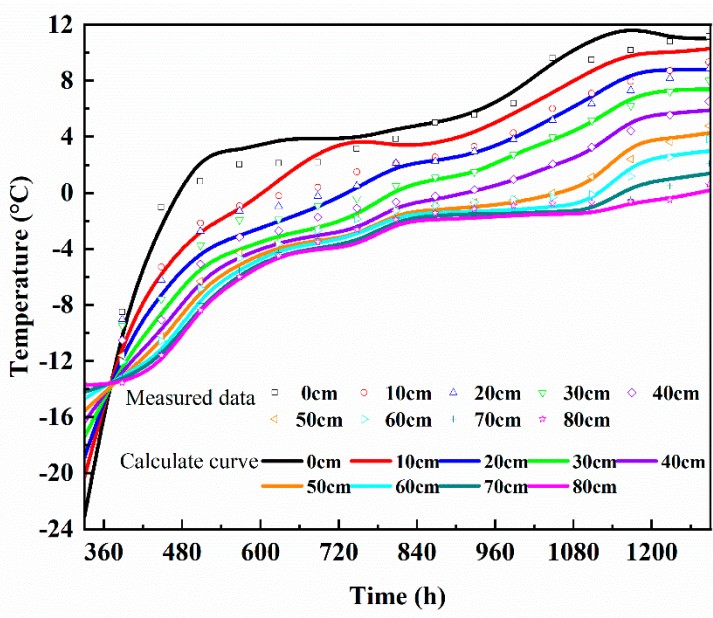

**Figure 10.** Measured and computed temperatures at different heights.

Based on the findings presented in Figure 11, it is evident that the numerical simulation of soil water content during the thawing stage exhibits some discrepancies when compared to the measured results. However, the overall result is acceptable. The formation of the maximum water content peak at a depth of 20–30 cm during the melting process can be attributed to several factors. Firstly, water in the upper melting zone migrates towards the freeze–thaw interface due to gravity and the temperature gradient. A portion of this water freezes into ice, while the remaining liquid water faces difficulty in further infiltration due to the obstructive effect of the aggregated ice layer. Consequently, the water from the melting zone accumulates near the freeze–thaw interface, resulting in the formation of a peak point of water content at this location. The disparity between the observed data and the model calculation can be attributed to two main reasons. Firstly, the soil test involves complex basic parameters and boundary conditions, whereas the model calculation tends to lean towards an idealized state and fails to account for all the conditions. Secondly, errors in the measurement of the test equipment prevent an accurate reflection of water migration in the soil under actual working conditions.

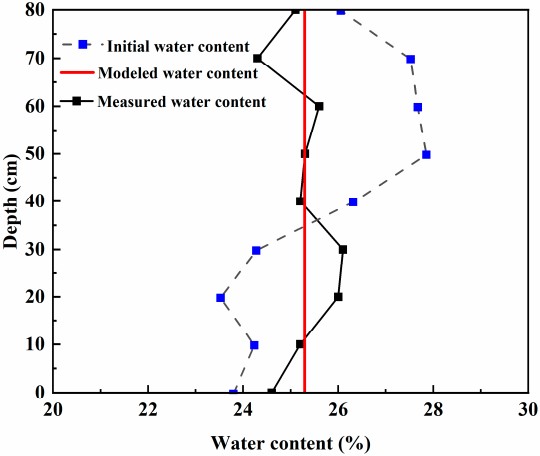

**Figure 11.** Computed profiles of water content.

## 5. Conclusions

In order to investigate the freezing and thawing mechanisms of farmland soil in cold regions and mitigate the occurrence of spring flooding disasters, this study focuses on Heilongjiang Province, a representative cold zone in northeast China. The research is centered on the frozen-thawed soil of farmland as the subject of investigation. By artificially simulating the climatic conditions of the cold region and utilizing a large-scale low-temperature laboratory, the evolution of temperature and moisture fields in farmland soil during the freezing and thawing process is examined. The study employs the principles of mass and energy conservation and introduces the concepts of relative saturation and solid–liquid ratio. Through mathematical transformations, a theoretical model of soil water–heat coupling is developed, with relative saturation and temperature serving as the field functions. The main findings of this research are as follows:

(1) The large-scale geotechnical model test is conducted using similarity theory, where the prototype project is scaled up and replicated in the laboratory. This allows for the simulation of the large-span and long-time cryogenic process of the engineering prototype within a shorter time frame.

(2) The cooling process of soil can be categorized into three phases: rapid cooling, slow cooling, and freezing stabilization. In the initial stage, the soil temperature decreases rapidly, with the main occurrence of violent water-ice phase transitions. As the soil depth increases, the volatility of the soil temperature gradually diminishes. During the freezing stage, the freezing line, represented by the $0\ ^{\circ}$C temperature contour, moves downward as the external ambient temperature decreases, indicating an increase in freezing depth. In the thawing stage, the temperature of the upper surface of the soil gradually rises with the increase in external ambient temperature, signifying an increase in thawing depth.

(3) Throughout the melting stage, the soil water content exhibits a gradual increase as the temperature rises. The range of variation in water content at depths of 30 cm, 40 cm, 50 cm, and 80 cm during the melting stage was found to be 0.12% to 0.52%, 0.47% to 1.08%, 0.46% to 1.96%, and 0.8% to 3.23%, respectively.

(4) By employing the principles of mass conservation, energy conservation, Darcy's law of unsaturated soil water flow, and the theory of heat conduction, a theoretical model of soil water–heat coupling was constructed. This model incorporates relative saturation and temperature as functions of the field and demonstrates a good match between the simulated temperature field and moisture field with the measured data. This indicates the effectiveness of the numerical model in revealing the freezing and thawing mechanisms of cold-region farmland soil.

**Author Contributions:** Conceptualization, M.H.; methodology, M.H.; software, M.W.; validation, S.G., M.H. and M.W.; formal analysis, A.S., M.H. and M.W.; investigation, A.S.; resources, S.G.; data curation, M.H.; writing—original draft preparation, M.H.; writing—review and editing, M.W.; visualization, C.L.; supervision, Y.G.; project administration, C.X.; funding acquisition, M.W. All authors have read and agreed to the published version of the manuscript.

**Funding:** This research was funded by the National Natural Science Foundation of China (U20A20318), Major Science and Technology Project of Ministry of Water Resources (SKS-2022095), Natural Science Foundation of Heilongjiang Province (ZD2023E007), Heilongjiang Provincial Research Institutes Scientific Research Business Fund Project (CZKYF2023-1-A009), General program of China Postdoctoral Fund (2021M690946), and the General Project of Heilongjiang Postdoctoral Fund (LBH-Z20094).

**Data Availability Statement:** All data generated or analyzed during this study are included in this article.

**Conflicts of Interest:** The authors declare no conflict of interest.

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
