# Peer review of "Large-Scale Freezing and Thawing Model Experiment and Analysis of Water–Heat Coupling Processes in Agricultural Soils in Cold Regions"

_water, doi:10.3390/w16010019_

Round 1

Reviewer 1 Report

Comments and Suggestions for Authors

Article of interest about the freezing and thawing of the soil in agricultural areas of a cereal-producing province of China. This process of soil freezing with very low temperatures, below 30ºC, significantly influences production and erosive phenomena due to rapid thaw. In their introduction, the authors adequately express the research on the topic. The objective for estimating soil temperature and moisture changes on agricultural land is correct.

In methodology, a similarity model is proposed experimentally in the laboratory, measuring temperature and humidity in the freeze-thaw cycles, using sun-dried clay soil samples.

However, the environmental data used to generate the model in the laboratory are not very clarifying. More information must be provided on soil temperatures and humidity during the 12 months of the year. I am concerned that the laboratory experiment does not exactly reproduce the external environment, and consequently the results will not be valid. For example, I do not see that an edaphic study has been carried out prior to the laboratory experiment.

Although I see the experimental design as correct, I insist, I do not see that there really is an artificial reproduction of the external environment.

Reviewer 2 Report

Comments and Suggestions for Authors

Thank you for providing the opportunity to review this manuscript which focuses on analyzing and modeling water—heat coupling process in cold region soils. Overall, I felt the manuscript was generally well written, unique, and well-structured. Most of my critical comments involve the need for additional details needed in the methods section.

Specific points:

Line 60: How does temperature facilitate water movement? Explain the theory. Typically, this would involve differences in height and pressure.

Table 1: Provide more clarity on what N, 1, 1 mean.

Figure 3: Provide more discussion of what is shown in Figure 3. What are the three different compartments?

Figure 5: In the figure caption, explain what the three different boxes are. Also, in the bottom figure, why are there no moisture sensors in the lower half of the figure?

Line 244: Show/explain better how it is evident that energy exchange and temperature changes are more pronounced at shallower depths.

Lines 251-252: I don’t see evidence of this in your results. Show evidence for this.

Page 8: What is the figure on bottom of this page? Figure 6? In (a), I don’t see how this is representative of a natural system. In natural systems, the surface layers cool faster than the subsurface. It appears to be reversed in your set up. Same comment as in figure (a) but reversed for heating.

Line 277: Do you mean “Measurement of sample water content”?

Figure 7: Shouldn’t it say, “Liquid water content at 30cm…Liquid water content at 40 cm…etc.”? Also, shouldn’t the legends say, “Initial water content” and “Final water content”?

Line 308: What is “drainage base soil”?

Line 438: Say “If we define the relative saturation…”

Line 489: I do not see a trend as you state here.

Line 496: What is “test data”, observed data?

Figure 11: In the legends suggest saying “Initial water content (measured)” and “Modeled water content” rather than calculated.

Comments on the Quality of English Language

English writing overall appears fine. Could use one more review. 

Reviewer 3 Report

Comments and Suggestions for Authors

Paper devoted to large-scale freezing and thawing model experiment and analysis of water-heat coupling processes in agricultural soils in cold regions. Heilongjiang Province, the largest commercial grain base in China, experiences significant challenges due to the environmental effects on its soil. The freezing and thawing cycle in this region leads to the transport of water and heat, as well as the exchange and transfer of energy. Consequently, this exacerbates the flooding disaster in spring and severely hampers farming activities such as plowing and sowing. To gain a better understanding of the freezing and thawing mechanism of farmland soil in cold regions and prevent spring flooding disasters, this study focuses on Heilongjiang Province as a representative area in northeast China. The research specifically investigates the frozen and thawed soil of farmland, using a large-scale low-temperature laboratory to simulate both artificial and natural climate conditions in the cold zone. By employing the similarity principle of geotechnical model testing, the study aims to efficiently simulate the engineering prototypes and replicate the process of large-span and long-time low temperature. The investigation primarily focuses on the evolution laws of key parameters, such as the temperature field and moisture field of farmland soil during the freeze-thaw cycle. The findings demonstrate that the cooling process of soil can be categorized into three stages: rapid cooling, slow cooling, and freezing stabilization. As the soil depth increases, the variability of soil temperature gradually diminishes. During the melting stage, the soil's water content exhibits a gradual increase as the temperature rises. The range of water content variation during thawing at depths of 30cm, 40cm, 50cm, and 80cm is 0.12% to 0.52%, 0.47% to 1.08%, 0.46% to 1.96%, and 0.8% to 3.23%, respectively. To analyze the hydrothermal coupling process of farmland soil during the freeze-thaw cycle, a theoretical model of hydrothermal coupling was developed based on principles of mass conservation, energy conservation, Darcy's law of unsaturated soil water flow, and heat conduction theory. Mathematical transformations were applied after defining the relative degree of saturation and solid-liquid ratio as field functions with respect to the relative degree of saturation and temperature. The simulated temperature and moisture fields align well with the measured data, indicating that the water-heat coupling model established in this study holds significant theoretical and practical value for accurately predicting soil temperature and moisture content during the spring sowing period, as well as for efficiently and effectively utilizing frozen soil resources in cold regions.

Topic of this paper is interesting and relevant but there are a comments:

1. In the Introduction, it is necessary to justify the choice of the region for the study.

2. In conclusion, it makes sense to note the main differences between the studied processes in Heilongjiang Province from other regions.
